https://doi.org/10.1038/s42003-021-01684-3　　**OPEN**

# Global mapping of protein–metabolite interactions in *Saccharomyces cerevisiae* reveals that Ser-Leu dipeptide regulates phosphoglycerate kinase activity

Marcin Luzarowski [1✉], Rubén Vicente[2], Andrei Kiselev[1,3], Mateusz Wagner[1,4], Dennis Schlossarek [1], Alexander Erban [1], Leonardo Perez de Souza [1], Dorothee Childs[5], Izabela Wojciechowska[1], Urszula Luzarowska [1,6], Michał Górka[1], Ewelina M. Sokołowska[1], Monika Kosmacz[1,7], Juan C. Moreno[1,7], Aleksandra Brzezińska[1], Bhavana Vegesna[1], Joachim Kopka [1], Alisdair R. Fernie [1], Lothar Willmitzer[1], Jennifer C. Ewald [8] & Aleksandra Skirycz [1,9✉]

Protein–metabolite interactions are of crucial importance for all cellular processes but remain understudied. Here, we applied a biochemical approach named PROMIS, to address the complexity of the protein–small molecule interactome in the model yeast *Saccharomyces cerevisiae*. By doing so, we provide a unique dataset, which can be queried for interactions between 74 small molecules and 3982 proteins using a user-friendly interface available at https://promis.mpimp-golm.mpg.de/yeastpmi/. By interpolating PROMIS with the list of predicted protein–metabolite interactions, we provided experimental validation for 225 binding events. Remarkably, of the 74 small molecules co-eluting with proteins, 36 were proteogenic dipeptides. Targeted analysis of a representative dipeptide, Ser-Leu, revealed numerous protein interactors comprising chaperones, proteasomal subunits, and metabolic enzymes. We could further demonstrate that Ser-Leu binding increases activity of a glycolytic enzyme phosphoglycerate kinase (*Pgk1*). Consistent with the binding analysis, Ser-Leu supplementation leads to the acute metabolic changes and delays timing of a diauxic shift. Supported by the dipeptide accumulation analysis our work attests to the role of Ser-Leu as a metabolic regulator at the interface of protein degradation and central metabolism.

A list of author affiliations appears at the end of the paper.

Metabolism is a complex system of chemical reactions that converts external nutrients to cellular building blocks and energy, as well as signalling molecules, defence agents and means of communication. In response to perturbations of nutrient supply or intracellular demands, metabolite concentrations and their conversion rates can change by orders of magnitude within seconds[1]. These timescales are too fast for transcriptional regulation, and thus cells have evolved more direct means of regulation: for example, metabolites themselves can act as regulators. Metabolites can regulate their pathways, balance competing pathways and coordinate metabolism with the physiology of the cell by interacting with and regulating proteins[2]. Examples of protein–metabolite interactions (PMIs) can be found in virtually all protein functional classes, ranging from metabolic enzymes to structural proteins to signalling components, such as transcription factors and kinases[3–7].

Regulation by PMI can be especially important for single-cell organisms that face constant changes in their environment and nutrient supply[2]. The yeast S. cerevisiae is a well-established single-cell model organism, and its metabolism has been extensively studied in the context of biotechnology, biomedicine and ecology. A recent study suggested that 29 out of 56 reactions in central yeast metabolism were at least partially regulated by allosteric interaction[8]. Identifying allosteric interactions can significantly improve the predictive power of metabolic models and the success of bioengineering approaches[9]. In addition to regulating the activity of metabolic enzymes, PMIs can also have global regulatory functions in coordinating metabolic fluxes with the physiology of the cell[10,11].

Despite their significant role in regulating metabolism and coordinating physiology, PMIs have remained understudied. For S. cerevisiae, there are approximately six times fewer reports of experimentally validated PMIs than protein–protein interactions (PPIs) in the STITCH database, which is a comprehensive resource integrating PMI for 430,000 chemicals[12,13]. Therefore, we expect to find many more regulatory functions of metabolites in the yeast cell. However, discovering these functions of metabolites requires suitable methods for globally capturing PMIs.

Powerful approaches that enable PMI studies at the cell-wide scale have been recently reported[14]. These technologies include affinity purification[15], thermal proteome profiling[16], drug affinity responsive target stability[17], small molecule limited proteolysis[18], tandem affinity purification[19,20] and capture compound mass spectrometry[21]. These are conceptually very different strategies, but they all share a common characteristic: namely, they require a predefined protein or metabolite as a bait. Consequently, they are ideal for studying interactions of a single metabolite or protein. However, they cannot capture the global overview of the interactome in an unbiased way.

To address this limitation, we have developed an approach, termed PROMIS, which enables a cell-wide analysis of the protein–metabolite and protein–protein interactomes[22,23]. Similar to the previously mentioned approaches, PROMIS starts with a native cellular lysate and thus operates in close to in vivo conditions. In brief, PROMIS combines size separation of complexes with proteomics and metabolomics analysis of the obtained fractions and exploits co-elution to define putative interactors. Thereby, needs neither a specific protein nor a specific metabolite as a bait. While this approach may not allow direct identification of binding partners, PROMIS is an ideal method of testing the complexity of an interactome and obtaining leads for targeted studies.

In the current study, we use PROMIS for systematic analysis of protein–small molecule interactions in Saccharomyces cerevisiae. We assayed interactions between 74 small molecules and 3982 proteins in the native cell lysate and recovered 16% of the previously reported binding events. We provide a unique data set of 225 interactions for 22 individual metabolites and explore specific examples of metabolite regulators. Most excitingly, our results point to the role of proteogenic dipeptides as metabolic regulators at the interface of protein degradation and central metabolism.

## Results

**PROMIS detects hundreds of candidate protein–metabolite interactions.** The goal of this work was to generate a proteome- and metabolome-wide map of protein–metabolite complexes of actively dividing and metabolically active S. cerevisiae. The diploid, prototrophic YSBN2 strain in the logarithmic phase of growth was used as starting material. The overall experimental strategy included: (i) preparation of the native, soluble lysate, (ii) size fractionation of complexes using a size exclusion chromatography (SEC) and (iii) untargeted analysis of the complex components using mass spectrometry-based metabolomics and proteomics[22,23] (Fig. 1a). In total, we collected 48 fractions from three biological replicates. Thirty-eight of the 48 fractions contained proteins and protein complexes spanning from 5.2 MDa to 20 kDa.

Metabolomics analysis identified 1016 small molecules of the mass between 100 and 1500 Da (Supplementary Data S1) that separated together with protein complexes and were, therefore, classified as protein-bound. A protein-free small-molecule extract was used as a negative control to exclude the unlikely possibility that free metabolites would elute together with the high-molecular-weight, protein-containing fractions. Indeed, the negative control tests confirmed this was not the case. Overall, 74 of the identified small molecules could be annotated to a specific compound using chemical standards, and included: purines and pyrimidines, amino acids, dipeptides and cofactors, as well as signalling molecules (3', 5'-cAMP), and transporters (carnitine) (Supplementary Data S2). Fifty of these 74 small molecules are known or were predicted to be a part of a yeast protein–metabolite complex (STITCH database), but binding was previously experimentally confirmed for only 15 of these small molecules in yeast.

Proteomic analysis identified 3982 proteins (Supplementary Data S3), which accounted for almost 90% of all yeast proteins expressed during the log phase of growth[24] and around 60% of the yeast proteome[25]. Twenty-seven per cent of the identified proteins were annotated as subunits of protein complexes (21% of the proteome), 7.5% were involved in molecular transport (8.3% of the proteome), 5% were kinases (3.6% of the proteome) and 8% had putative or unknown functions (17% of the proteome). Proteins integral to membranes or associated with the plasma membrane were significantly underrepresented (0.71-fold enrichment), whereas cytoplasmic and nucleolar proteins were over-represented (1.28 and 1.48-fold enrichment, respectively) (PANTHER database, Supplementary Data S4).

Given that the majority of the proteins and metabolites had complex elution patterns characterised by more than one elution maximum, we split the data profiles into single peaks; this is referred to as deconvolution[26]. By doing so, we obtained 1320 and 125 peaks for unknown and annotated metabolites, respectively, and 5834 protein peaks. These were used for further analysis (Supplementary Data S5–S7).

We also determined whether the protein–protein complexes remained intact during the PROMIS separation by examining 5834 protein peaks and calculating the apparent mass of a protein complex based on its elution maximum. We then calculated the ratio between the apparent mass and the theoretical monomeric mass of a protein. This ratio reflects the oligomerisation state of a

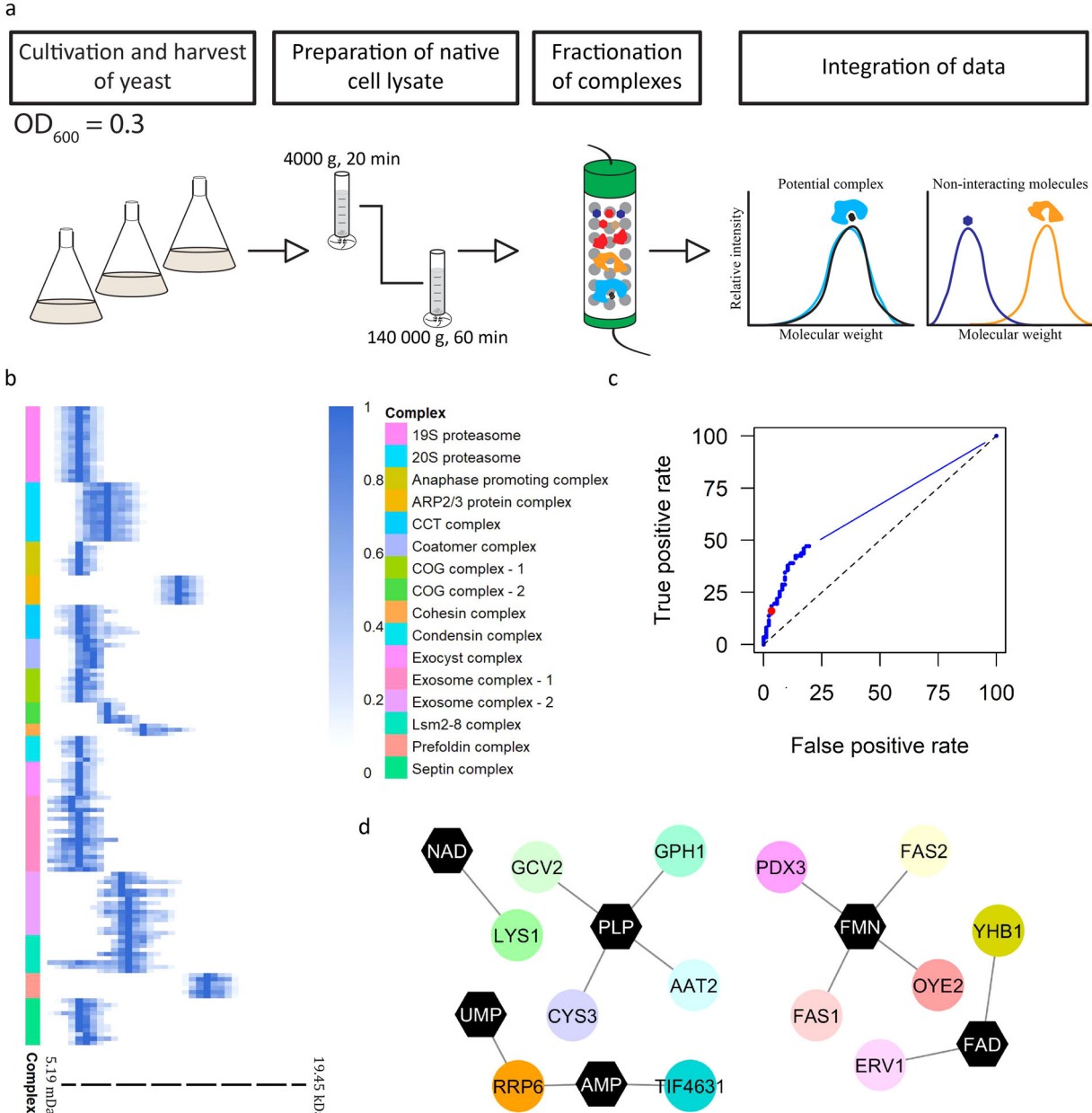

**Fig. 1 PROMIS allows for system-wide detection of protein–small molecule and protein–protein complexes using size exclusion chromatography. a** Dividing yeast cells were harvested in the logarithmic phase of growth and were used as a source of endogenous protein–protein and protein–metabolite complexes. Complexes were fractionated using size exclusion chromatography, lyophilised and subjected to methyl tert-butyl ether-methanol-water extraction. Polar metabolites and proteins were analysed by liquid chromatography-mass spectrometry. **b** Known yeast protein macro-complexes remain intact. Multiple subunits of known protein macro-complexes co-elute together. The peak elution profiles of the components of 14 known protein macro-complexes are depicted (Supplementary Data S7 and S8). The intensity was calculated relative to the maximum intensity of a given protein measured across size exclusion chromatography separation range. Distinct colours are used to mark different protein macro-complexes. **c** The receiver operating characteristic curve represents a trade-off between numbers of captured true-positive and false-positive protein–metabolite interactions by varying the Pearson correlation coefficient (PCC) (Supplementary Data S10). The red dot indicates the chosen threshold (PCC ≥ 0.7) used for determining complexes. **d** Interaction network of captured known protein–metabolite complexes. Overall, 14 of the 87 known protein–metabolite interactions were re-captured in the PROMIS experiment (Supplementary Data S9).

protein (referred to as its oligomeric state ratio). Assuming that an oligomeric state ratio above 1.5 would indicate an interaction with another protein, we identified 4981 protein peaks, corresponding to 3408 proteins, as part of a multimeric complex (Supplementary Data S8). We further validated the effectiveness of fractionation by examining the elution profiles of 14 known protein macro-complexes, such as the proteasome[27]. As anticipated, the respective components of the analysed complexes

shared an elution profile, validating the suitability of PROMIS for the isolation, fractionation and identification of native protein complexes (Fig. 1b, Supplementary Data S7).

We estimated co-elution, which we use to define putative interactors, by calculating the Pearson correlation coefficient (PCC) between all annotated metabolite and all protein peaks present in our data set. We then determined the influence of using the PCC threshold on the number of detected true PMIs,

which we retrieved from the STITCH database. For this purpose, we created the list of reported true interactions comprising 87 PMIs, including only proteins and metabolites identified in our data. Due to the lack of experimental evidence confirming that given protein–metabolite pair does not interact, we constructed a list of false positives by calculating the PCC of 87 randomly picked protein–metabolite pairs present in our dataset (100 iterations). Next, we compared PCC values obtained for true interactions retrieved from the STITCH database (Supplementary Data S9) with randomly picked values (for a more detailed description see Supplementary Methods). We calculated the receiver operating characteristic curve, which showed the trade-off between specificity (a low number of false-positive hits) and sensitivity (the number of retrieved true–positive interactions) (Fig. 1c) (Supplementary Data S9 and S10). To assure specificity but to not reduce sensitivity, we applied a PCC threshold of 0.7 to determine the PMIs (false discovery rate = 17.6%). We recovered 14 of the 87 true PMIs, which is five times greater than the number expected by chance and achieving a true-positive rate comparable to a recent MS-based proteome-wide PMI study in *Escherichia coli*, reporting protein binders for 20 different metabolites (Fig. 1d)[18]. Moreover, the correlation coefficients calculated for the true protein–metabolite pairs were higher than the permuted values (Supplementary Fig. S1).

It is important to note that when interpreting PROMIS results, PCC shouldn't be used to rank the interactions. However, we anticipate that many of the small molecules will have few specific protein partners, and so a single protein peak is expected to correspond to a single metabolite peak, equally there will be metabolites for which a single elution peak will correspond to the multiple protein partners, obscuring the PCC. In other words and in the latter case it is the co-elution alone, rather than PCC that is indicative of the interaction. Equally, because metabolite binding may vary depending on a protein oligomeric state or presence in a particular protein complex, multiple correlated protein–metabolite peaks will not always reflect confidence. Taken together, PROMIS results should be seen more as qualitative rather than quantitative, and we recommend that the choice of a PCC threshold should be governed by the best compromise between specificity and sensitivity estimated from the receiver operating characteristic curve (Fig. 1c, Supplementary Data S10).

Finally, we created a user-friendly interface, which can be mined for elution profiles of all measured metabolites and proteins, and for the PMIs. The interface is available at https://promis.mpimp-golm.mpg.de/yeastpmi/.

**The PROMIS data set captures 225 of the previously predicted yeast PMIs.** In addition to known PMIs, the STITCH database can be mined for predicted PMIs, where prediction is made based on the binding data available for the orthologous proteins, and assuming evolutionary conservation of the interactions[27,28]. We queried lists of predicted PMIs against the yeast PROMIS data set to provide experimental validation for the previously predicted complexes.

Of the 1122 predicted PMIs, we found experimental evidence for 225 interactions, engaging 22 unique metabolites (Fig. 2a, Supplementary Data S11). A majority of these interactions were between nucleoside monophosphates (NMPs)—such as (deoxy)-AMP, (deoxy)-GMP and UMP—and DNA-binding and RNA-binding proteins (Fig. 2b and Supplementary Fig. S2). After the NMPs, the second largest group was comprised of interactions between enzymes and cofactors (e.g. FMN, FAD, NAD(H) and PLP). Most notably, our dataset validated 19 of the predicted PLP binders, 14 of these were enzymes associated with amino acid metabolism.

In the next step, we decided to explore the list of 225 validated PMIs for those of potential regulatory nature. Herein, we will highlight a representative example, which we followed up and validated experimentally. Purines and pyrimidines are pivotal for multiple cellular processes. Perturbation of their homoeostasis leads to metabolic dysfunctions and has a serious impact on yeast growth[29–32]. Considering the importance of purine metabolism, we were intrigued by the interaction between xanthine and purine nucleoside phosphorylase (Pnp1), present in the list of 225 PMIs validated by PROMIS.

Pnp1 catalyses the conversion of guanosine and inosine to guanine and hypoxanthine, respectively. In the PROMIS data set, Pnp1 (monomeric mass 33 kDa) separated as two distinct elution peaks with maximum intensity in fractions corresponding to 138 kDa and 88 kDa. This indicates that, in vivo, Pnp1 exists in two different oligomeric forms or is part of a protein complex. Pnp1 co-eluted with its known substrate, inosine (Fig. 3a, b). In addition, Pnp1 co-fractionated with xanthine (PCC > 0.95) (Fig. 3c). As other enzyme–metabolite pairs identified in this work, the interaction between Pnp1 and xanthine represents a potential catalytic interaction. However, similar to human Pnp1, *Sc*Pnp1 is unable to metabolise adenosine and xanthosine. Thus, Pnp1–xanthine binding is more likely a putative regulatory interaction[33].

To test this hypothesis, we investigated whether xanthine affects Pnp1 activity. To this end, we purified recombinant Pnp1 from *S. cerevisiae* and used it in an enzymatic assay that measures the conversion rate of inosine to hypoxanthine[34]. The amount of hypoxanthine produced was measured over time using an LC-ESI-MS assay in the presence or absence of 100 μM xanthine. The addition of 100 μM xanthine lowered the total Pnp1 activity by up to 32% (Fig. 3d). The accumulation of xanthine in yeast may, therefore, lower Pnp1 activity and slow the conversion of inosine and guanosine to hypoxanthine and guanine, respectively, which subsequently would lead to the reduction of hypoxanthine and guanine level in yeast cells (Fig. 3e).

**The dipeptide Ser-Leu interactome comprises numerous proteins involved in protein and amino acid metabolism.** Of the 74 annotated metabolites that co-eluted with proteins, 36 were proteogenic dipeptides. In yeast, the sole reported dipeptide–protein interaction is between dipeptides with the basic N-terminal residue (Arg, Lys or His) and site-1, and between dipeptides with the bulky hydrophobic N-terminal residue and site-2 (Trp, Phe, Tyr, Leu or Ile) of the ubiquitin ligase, Ubr1[35]. Importantly, two of the type 1 dipeptides (Arg-Phe and Lys-Phe) also co-elute with Ubr1 in our data set (Supplementary Fig. S3). Encouraged that we could recapitulate known binding, we decided to determine the precise identity of the protein interactors of a single selected and representative dipeptide, namely Serinyl-Leucine (Ser-Leu). The Ser-Leu elution profile spans reproducibly across a PROMIS protein separation range in all three replicates and is characterised by three local maxima, indicating the presence of a multitude of protein partners. The three Ser-Leu peaks co-elute (PCC ≥ 0.7) with 239, 376 and 182 proteins.

To validate the predicted partners, we performed affinity purification experiments starting with Ser-Leu as the bait (c.f. 'Methods'). We used agarose beads coupled to Ser-Leu by the $NH_2$ group of serine (N-Ser-Leu) or the COOH group of leucine (Ser-Leu-C). We found 162 proteins that were significantly enriched in eluates from the N-Ser-Leu and Ser-Leu-C beads, constituting putative Ser-Leu targets (Supplementary Fig. S4 and Supplementary Data S12). Proteins involved in protein metabolism (amino acid biosynthesis, protein folding, proteasome, proteins involved in translation and protein targeting) were

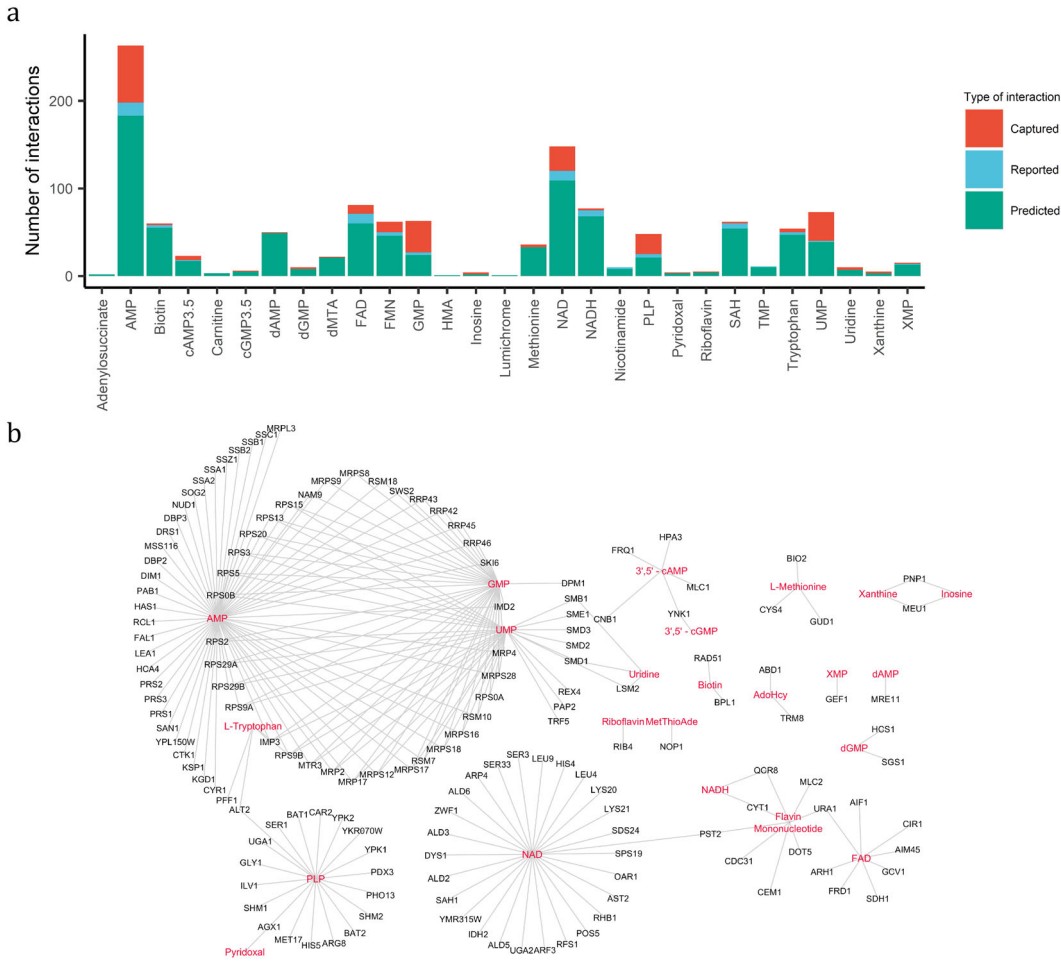

**Fig. 2 PROMIS provides experimental validation for multiple predicted protein–small molecule complexes. a** Number of captured protein–small molecules interaction in relation to previously reported and predicted interactions for each metabolite validated by our data set (Supplementary Data S9 and S11). **b** The interaction network of 225 STITCH predicted protein–small molecule interactions validated in this study (Supplementary Data S11). Edges represent protein–small molecule interactions and were imported from STITCH, based on the experimental evidence (score ≥ 0.4). Proteins and metabolites are marked as black and red, respectively. Metabolite abbreviations: AdoHcy adenosyl homocysteine, AMP adenosine monophosphate, dMTA/MetThioAde methylthioadenosine, HMA hydroxy methylglutaric acid, PLP pyridoxal phosphate, SAH S-adenosyl-homocysteine, TMP thymidine monophosphate, UMP uridine monophosphate, XMP xanthine monophosphate.

significantly overrepresented (false discovery rate < 0.05) (Supplementary Fig. S5).

To complement the affinity purification experiments, we used an independent biochemical method for the identification of protein partners of small-molecule ligands, namely thermal proteome profiling. Thermal proteome profiling monitors changes in protein thermal stability caused by ligand binding[16]. We analysed our obtained data by applying the non-parametric analysis of response curves method[36]. The method is independent of melting temperature estimation and tests the differences in curves rather than the differences in melting temperature. We found 94 potential targets that had melting profiles significantly affected by Ser-Leu treatment (Benjamini-Hochberg $P$ value ≤ 0.05) (Supplementary Data S13 and Supplementary Fig. S6). Again, proteins involved in protein metabolism were significantly enriched (false discovery rate < 0.05) (Supplementary Fig. S7).

In total, 86 proteins, assigned as Ser-Leu-binding proteins based on at least two of the three experimental strategies, were queried against a STRING database (Fig. 4a). Seventy-seven of the 86 proteins were part of the resulting PPI network (Fig. 4b and Supplementary Fig. S8). Functional and enrichment analyses showed a significant overrepresentation of proteins involved in

amino acid biosynthesis, translation, protein folding, degradation and targeting (Supplementary Fig. S9). Five proteins, identified by all three independent approaches (PROMIS, affinity purification and thermal proteome profiling), were assigned as high-confidence Ser-Leu-binding proteins (Fig. 4a). These five proteins were two subunits of the T-complex (Cct3 and Cct8)[37], the regulatory subunit of acetolactate synthase complex (Ilv6)[38], polyamine acetyltransferase (Paa1)[39] and the yeast prion protein (New1)[40].

Particularly intriguing was the appearance of Ilv6, which is involved in the biosynthesis of branched-chain amino acids (valine, leucine and isoleucine) and feedback inhibited by the binding of valine[38]. Prompted by the published data, we investigated the elution profiles of Ilv6, catalytic subunit of acetolactate synthase complex (Ilv2) and dipeptides containing branched-chain amino acids. We found that, in addition to Ser-Leu, Val-Leu, Leu-Leu, Thr-Leu, Ile-Leu, Asn-Ile and Thr-Val also co-migrated with the subunits of acetolactate synthase complex (Fig. 4c).

**Ser-Leu is a regulator of the glycolytic enzyme Pgk1.** The Ser-Leu elution profile is characterised by three local maxima,

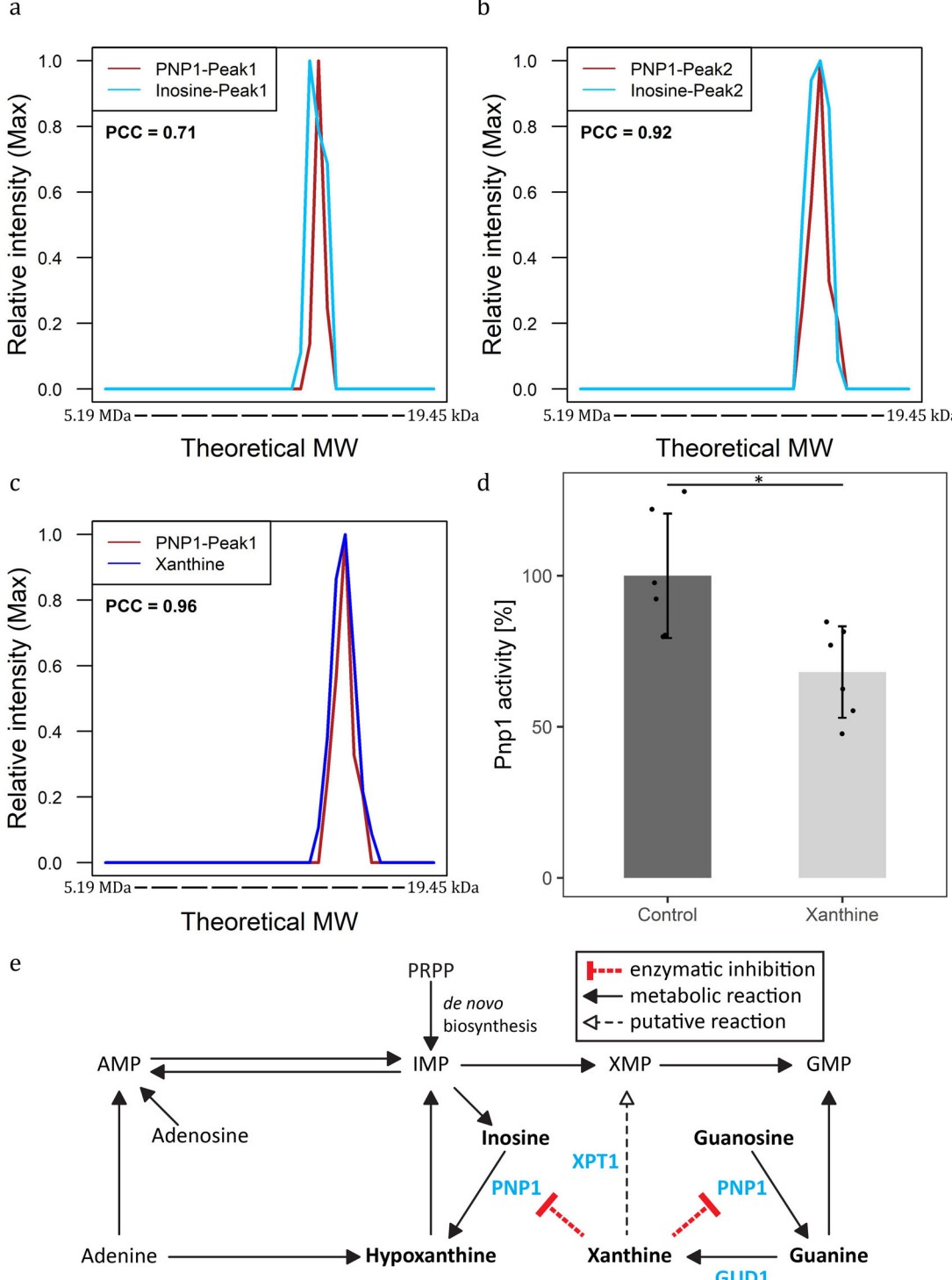

**Fig. 3 Functional validation of the Pnp1–xanthine interaction. a–c** Elution profiles of Pnp1, with its known substrate inosine (**a,b**) and putative ligand xanthine (**c**) (Supplementary Data S6 and S7). The intensity was calculated relative to the maximum intensity of the molecule measured across size exclusion chromatography fractions. The theoretical molecular weight (MW) was calculated using reference proteins. **d** Xanthine inhibits Pnp1 activity. Total activity of recombinant Pnp1 in the presence of 100 μM xanthine was measured using an liquid chromatography-mass spectrometry-based assay (Supplementary Data S18). Inhibition was calculated in relation to Pnp1 activity in the absence of xanthine. Data represent the means ± SD, $n = 6$ independent samples. Asterisks denote significant difference (non-paired, two-tailed $t$ test $P$ value < 0.05). **e** Scheme of purine degradation pathway with predicted regulatory interaction between Pnp1 and xanthine. Molecules discussed in this study are depicted in bold. Enzymes are additionally marked in blue.

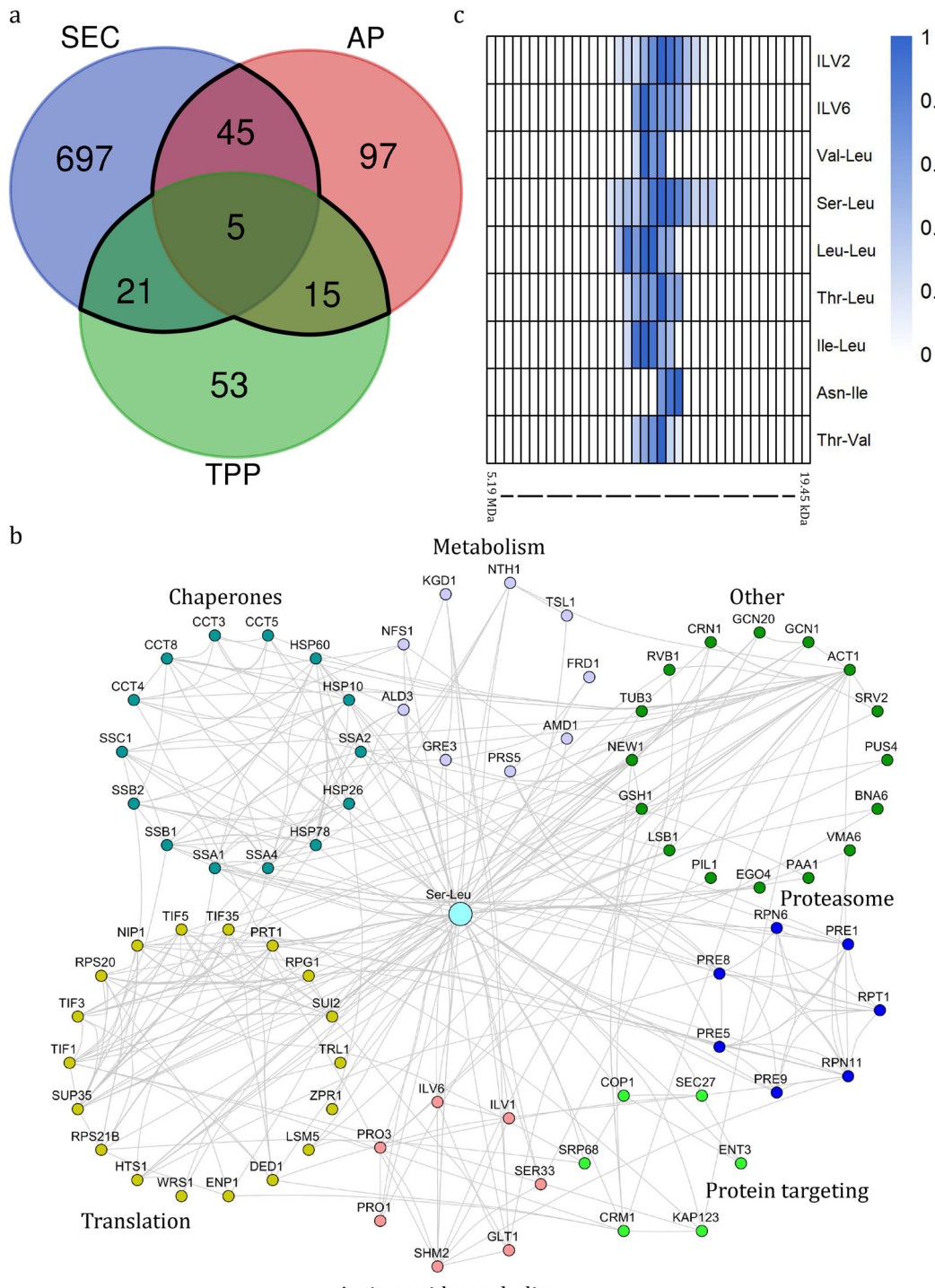

**Fig. 4 Characterisation of Ser-Leu interactome. a** Venn diagram showing the number of putative Ser-Leu targets identified using size exclusion chromatography (SEC), affinity purification (AP) and thermal proteome profiling (TPP) (Supplementary Data S12, S13 and S17). An overlap between at least two orthogonal approaches (86 proteins) was considered to represent the Ser-Leu interactome and is marked in black. **b** The Ser-Leu interactome network. Edges represent protein–protein interactions and were imported from STRING, based on the experimental evidence (score ≥ 0.4). Functionally related proteins are grouped together. Distinct colours are used to mark different protein groups. **c** Heatmap showing co-elution of catalytic (Ilv2) and regulatory (Ilv6) subunits of the acetolactate synthase complex with dipeptides containing branched-chain amino acids (Supplementary Data S6 and S7). The intensity was calculated relative to the maximum intensity measured across the SEC fractions. The theoretical molecular weight was calculated using reference proteins.

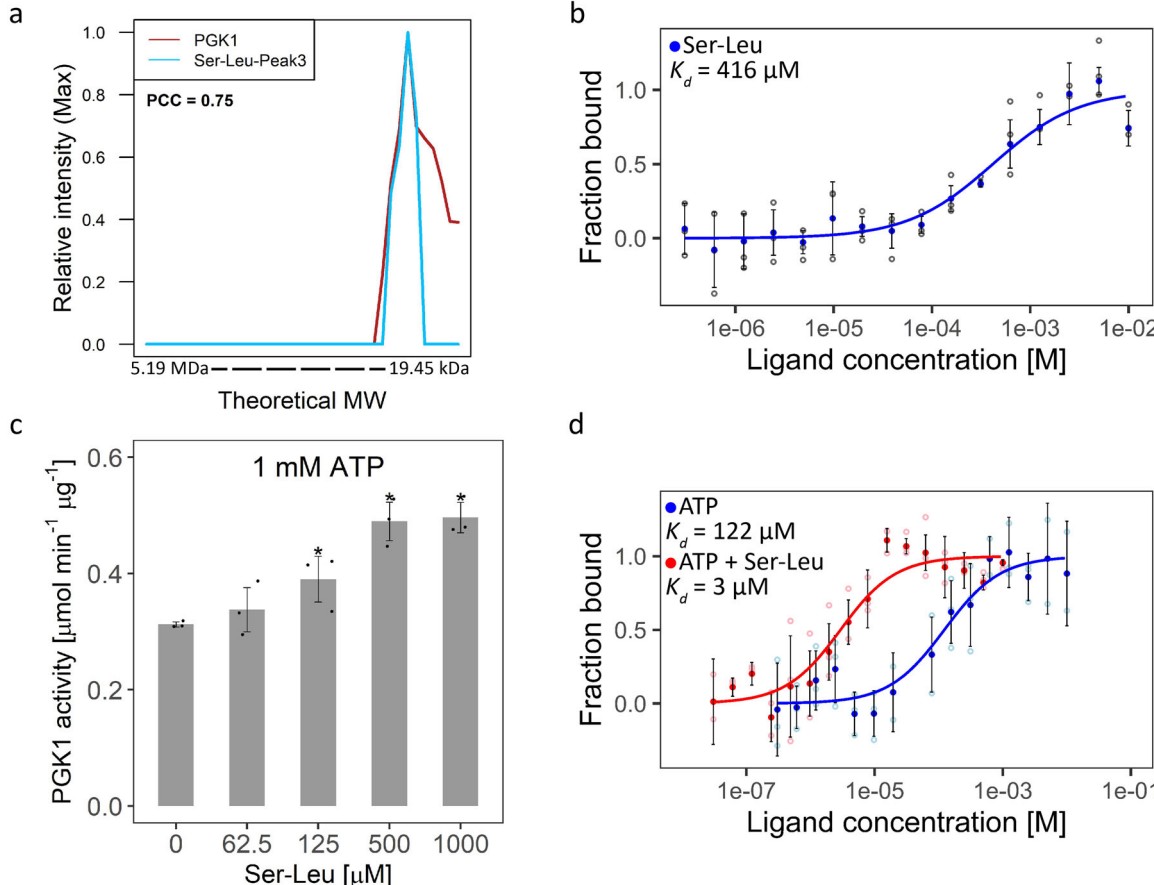

**Fig. 5 Characterisation of the Pgk1–Ser-Leu interaction. a** The elution profile of Pgk1 and its putative ligand Ser-Leu (Supplementary Data S6 and S7). The intensity was calculated relative to the maximum intensity of the molecule measured across size exclusion chromatography fractions. The theoretical molecular weight (MW) was estimated using reference proteins. The Pearson correlation coefficient (PCC) indicates a correlation coefficient calculated between depicted elution profiles. **b** Microscale thermophoresis analysis of Pgk1 and Ser-Leu binding (Supplementary Data S26). $K_d$ indicates dissociation constant. Data represent the means ± SD, $n = 3$ independent samples. **c** Functional validation of the interaction between Ser-Leu and Pgk1 (Supplementary Data S28). Ser-Leu significantly increases Pgk1 activity. Data represent the means ± SD, $n = 3$ independent samples. Asterisks denote significant difference (non-paired, two-tailed $t$ test $P$ value < 0.05). **d** Microscale thermophoresis analysis of Pgk1 and ATP binding in the presence of saturating concentrations (4 mM) of Ser-Leu (Supplementary Data S27). $K_d$ indicates dissociation constant. Data represent the means ± SD, $n = 3$ independent samples.

indicating co-presence of interacting proteins in respective fractions. However, when we checked the PROMIS data set, we found that all five of the high-confidence protein targets (identified simultaneously by affinity purification, thermal proteome profiling and PROMIS) corresponded to either the first or second Ser-Leu peaks, but none co-fractionated with the third peak. Prompted by our earlier observation that Tyr-Asp binds to plant glyceraldehyde-3-P dehydrogenase (GAPDH)[22], we searched for glycolytic enzymes among the 182 proteins co-eluting with the third peak of Ser-Leu and identified phosphoglycerate kinase (Pgk1) as a putative target of Ser-Leu (Fig. 5a).

We validated the direct interaction between Pgk1 and Ser-Leu using microscale thermophoresis with a determined $K_d$ of 416 µM (Fig. 5b)[41]. In comparison, no interaction could be measured between Pgk1 and Tyr-Asp, (Supplementary Fig. S10) and between Pgk1 and serine, which was used as a negative control (Supplementary Fig. S10). We decided for serine, as analysis of dipeptide uptake in yeast showed that an amino acid residue at the N-terminus has a more significant role in dipeptide recognition than one on the C-terminus[42]. In line with our results, recent systematic analysis of PMIs in central metabolism using nuclear magnetic resonance showed that Pgk in *Escherichia coli* does not bind to either serine or leucine[43].

We characterised the effect of the interaction between Pgk1 and Ser-Leu by testing whether Ser-Leu affects the activity of recombinant Pgk1[44]. We used a stopped enzymatic assay (Supplementary Fig. S11), which measures the conversion of 3-phosphoglycerate (3PGA) to bisphosphoglycerate (BPGA) and subsequently to glyceraldehyde-3-P (GAP), dihydroxyacetone-P (DAP) and finally glycerol-3-P (G3P)[45,46]. Micromolar concentrations of Ser-Leu significantly increased the activity of Pgk1; however, the activating effect was observable only at relatively low concentrations of the ATP used in the assay (below $V_{max}$) (Fig. 5c and Supplementary Fig. S12). Since high concentrations of ATP diminished the activating effect of Ser-Leu, we hypothesized that Ser-Leu may increase the affinity of Pgk1 towards ATP. To test this assumption, we used microscale thermophoresis to determine the $K_d$ of the interaction between Pgk1 and ATP in the presence of a saturating concentration of Ser-Leu (Fig. 5d). We first validated the interaction between Pgk1 and ATP ($K_d$ of 122 µM). Next, we demonstrated that Ser-Leu lowered $K_d$ of ATP binding by 40-fold, effectively increasing the affinity of Pgk1 for ATP ($K_d$ of 3 µM).

**Dipeptide accumulation is associated with glucose depletion.** To learn more about the biological context of Ser-Leu action, we decided to investigate dipeptide and amino acid accumulation

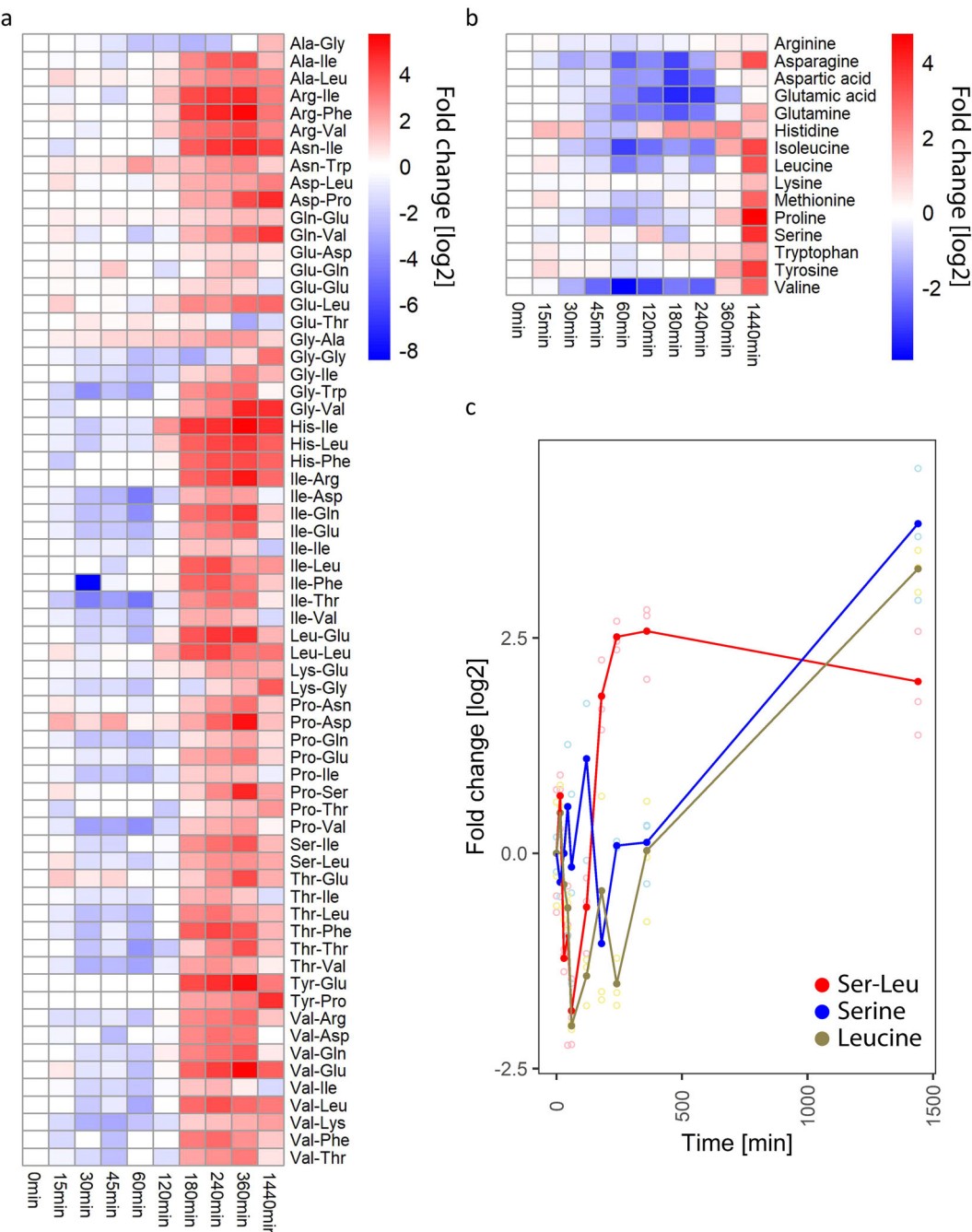

**Fig. 6 Analysis of dipeptide and amino acids fluctuations in yeast under control growth conditions (30 °C). a** Heatmap showing fluctuation of dipeptides level in yeast (Supplementary Data S30). **b** Heatmap showing fluctuation of amino acids level in yeast (Supplementary Data S30). **c** Plot showing fluctuation of Ser-Leu, serine and leucine level in yeast. Shown are relative changes to time point 0. Data represent the means $n = 3$ independent samples. Ratios were log transformed (log base 2).

during growth on glucose at optimal conditions (30 °C). For this purpose, yeast culture was grown to the stationary phase, followed by transfer to a fresh pre-treated medium (see 'Methods' section). Samples were harvested immediately after transfer to fresh medium and at multiple time points ranging from 15 to 1440 min, quenched in methanol and analysed by liquid chromatography-mass spectrometry. With few exceptions, all of the measured dipeptides accumulated after 180 min of growth (Fig. 6a), which corresponds to glucose depletion (Supplementary Fig. S13a). When compared with dipeptides, amino acids displayed a different accumulation pattern, characterised by an increase after 360 and 1440 min of growth (Fig. 6b).

More specifically, level of Ser-Leu and leucine decreased after 30 min of growth and started to accumulate after 180 and 1440 min of cultivation, respectively (Fig. 6c). In contrast, level of serine undergoes fewer fluctuations. Similarly to leucine, it accumulates after 1440 min of growth.

**Ser-Leu feeding affects both central metabolism and yeast growth**. In addition to Pgk1, Ser-Leu protein interactome comprised numerous other enzymes, from amino acids biosynthesis (Prs5, Ser33, Shm2, Ilv6, Glt1, Pro1, Pro3), the TCA cycle (Kgd1), purine (Amd1) and NAD metabolism (Bna6). To examine whether Ser-Leu binding translates into a metabolic effect we

followed changes in relative metabolite levels (here described as total intensity) and redistribution of carbon isotope (enrichment level [%] multiplied by relative metabolite level, here described as 13 C fraction intensity) in yeast cells upon Ser-Leu supplementation. Specifically, yeast cells at stationary phase were fed with $^{13}$C glucose together with either mock, 100 μM Ser-Leu or a mix of 100 μM serine and 100 μM leucine. Samples were harvested at multiple time-points ranging from 5 to 240 min following treatment, quenched in methanol and analysed by gas chromatography- and liquid chromatography-mass spectrometry (Supplementary Figs. S13–S21). Over time, the $^{13}$C glucose is taken up and metabolized by the cell and metabolites become enriched for $^{13}$C until the steady-state enrichment is reached. While changes in metabolites levels are valuable information to describe the metabolic state of an organism, they are limited in providing information regarding the flow of mass through the system. $^{13}$C enrichment provides further information to access the conversion rates of labelled substrates through metabolism, which can be used to estimate the production rate of a given metabolite[47].

The choice of Ser-Leu concentration was guided by the absolute cellular levels of Ser-Leu, which we estimated to approximate 6 μM by spiking different amounts of Ser-Leu (from 100 nM to 100 μM) into metabolic extract prepared from $^{13}$C labelled S288c yeast culture corresponding to the stationary phase of growth.

GC- and LC-MS analysis of the Ser-Leu, serine and leucine concentrations in the Ser-Leu supplemented cells revealed rapid Ser-Leu accumulation, which remained constant over time (Supplementary Fig. S13b). Neither serine nor leucine accumulated, at least during the duration of the Ser-Leu treatment, arguing that Ser-Leu was not degraded to its constituent amino acids (Supplementary Fig. S13cd).

Most conspicuously, Ser-Leu treatment led to a stark increase in the de novo production rate of the 3PGA (3-fold change), directly downstream of the Pgk1 activity, followed by an increase in pyruvic acid production (Fig. 7 and Supplementary Fig. S14). Interestingly, the excess of glycolytic 3PGA and pyruvate was directed away from the tricarboxylic acid cycle, as de novo production rates of all of the measured tricarboxylic acid cycle intermediates, citric acid, succinic acid, fumaric acid and malic acid were decreased (Supplementary Fig. S15). Similarly, and possibly as a consequence, also de novo synthesis of tricarboxylic acid cycle-derived amino acids: methionine, saccharopine (intermediate in the metabolism of lysine), proline, arginine and aspartate were downregulated (Supplementary Figs. S16–S18). Moreover, Ser-Leu treatment (i) led to upregulation of de novo synthesis of 5'-GMP, and accumulation of 3'-AMP, and adenine all being intermediates of purine metabolism (Supplementary Fig. S19), (ii) increased levels of two intermediates of sphingolipid metabolism, sphingosine and hydroxypalmitic acid (Supplementary Fig. S20) and (iii) elevated de novo synthesis of cofactors NADP$^+$ and FAD$^+$ (Supplementary Fig. S21). Similarly, to Ser-Leu, also amino acid feeding resulted in a number of metabolic changes. However, the observed effects were different; for instance, in contrast to the Ser-Leu, the amino-acid treatment did not affect de novo synthesis of the tricarboxylic acid cycle intermediates succinic acid, fumaric acid and malic acid.

Finally, and to complement our metabolic analysis we tested whether Ser-Leu supplementation affects yeast growth. For this purpose, starved yeast culture in stationary phase was supplemented with glucose together with either mock, 1 mM Ser-Leu or mixture of 1 mM serine and 1 mM leucine (Fig. 8a). Yeast growth was monitored by measuring OD$_{600nm}$ using an automatically recording incubator. Ser-Leu treatment affected yeast growth during early exponential phase and supplemented culture reached

higher OD$_{600nm}$ than mock. Ser-Leu treatment delayed diauxic shift for 30 min, therefore prolonged fermentation, and shortened respiration phase (Fig. 8b). In comparison to treatment with dipeptide, supplementation with a mixture of serine and leucine affected yeast growth much later (4 and 6 h upon treatment with dipeptide and amino acids, respectively) and did not delay the diauxic shift (Fig. 8c).

## Discussion

Herein, we used PROMIS to chart a map of protein–small molecule interactions (PMIs) in the model yeast *Saccharomyces cerevisiae*.

As a result, we report a unique data set resulting from an analysis of endogenous protein–metabolite and protein–protein complexes. Our most remarkable observation relates to the wealth of small molecules present in the protein complexes; this attests to the complexity of the protein–small molecule interactome and highlights an important but severely understudied role of small molecules as protein regulators. We report 225 previously predicted PMIs that could be validated using PROMIS. Considering that the STITCH database contains 87 true interactions for the same subset of metabolites and proteins, then a single PROMIS experiment was sufficient to nearly quadruple the number (from 87 to 312). We successfully queried the list of 225 validated interactions for binding events with a putative regulatory role, such as between Pnp1 and xanthine. However, in vivo significance of the xanthine inhibition of Pnp1 activity remains to be tested, xanthine binding to Pnp1 is an excellent example where querying a single PROMIS dataset is sufficient to retrieve regulatory interactions.

In addition to the previously predicted PMIs, the presented PROMIS data set can be mined for new binding events, assisting the discovery and functional characterisation of small-molecule regulators. In line with an analogous PROMIS study in Arabidopsis proteogenic dipeptides stood out as a major group of protein-bound small molecules[22]. A role of dipeptides in the regulation of central metabolism has been discussed before. Increase of proteogenic dipeptides in tumour-associated cells correlated with the glycolytic capacity of the tumour[48]. In comparison, treatment with the non-proteogenic dipeptide carnosine (β-alanyl-L-histidine) reduced the proliferative capacity of human gastric cancer cells by inhibiting glycolysis, mitochondrial oxidative phosphorylation and respiration[49]. Finally, the acidic dipeptide Tyr-Asp was found among small-molecule ligands of a glycolytic enzyme, GAPDH[22]. Here, we could demonstrate that Ser-Leu affects glycolysis via direct binding and activation of Pgk1. Consistent with the in vitro results, Ser-Leu feeding led to a rapid accumulation of an important glycolytic intermediate 3PGA. 3PGA is eventually converted into pyruvate but can also be re-directed into serine biosynthesis. While serine is an entry point into one-carbon metabolism, pyruvate is utilized to produce energy via either the tricarboxylic acid cycle (respiration) or the ethanol production (fermentation). Reduced levels of the tricarboxylic acid cycle intermediates, and tricarboxylic acid cycle-derived amino-acids, measured in response to the Ser-Leu supplementation point to pyruvate being directed away from the respiration, most likely into fermentation. These data are in line with the measured growth effects. Ser-Leu treatment delayed the diauxic shift, which is indicative of Ser-Leu supporting fermentation over respiration. Moreover, and since Ser-Leu accumulation accompanies glucose depletion characteristic for the late logarithmic phase of growth, we propose that Ser-Leu, and possibly also other proteogenic dipeptides, reinforce sugar repression of the tricarboxylic acid cycle in yeast cells when the glucose levels fall low.

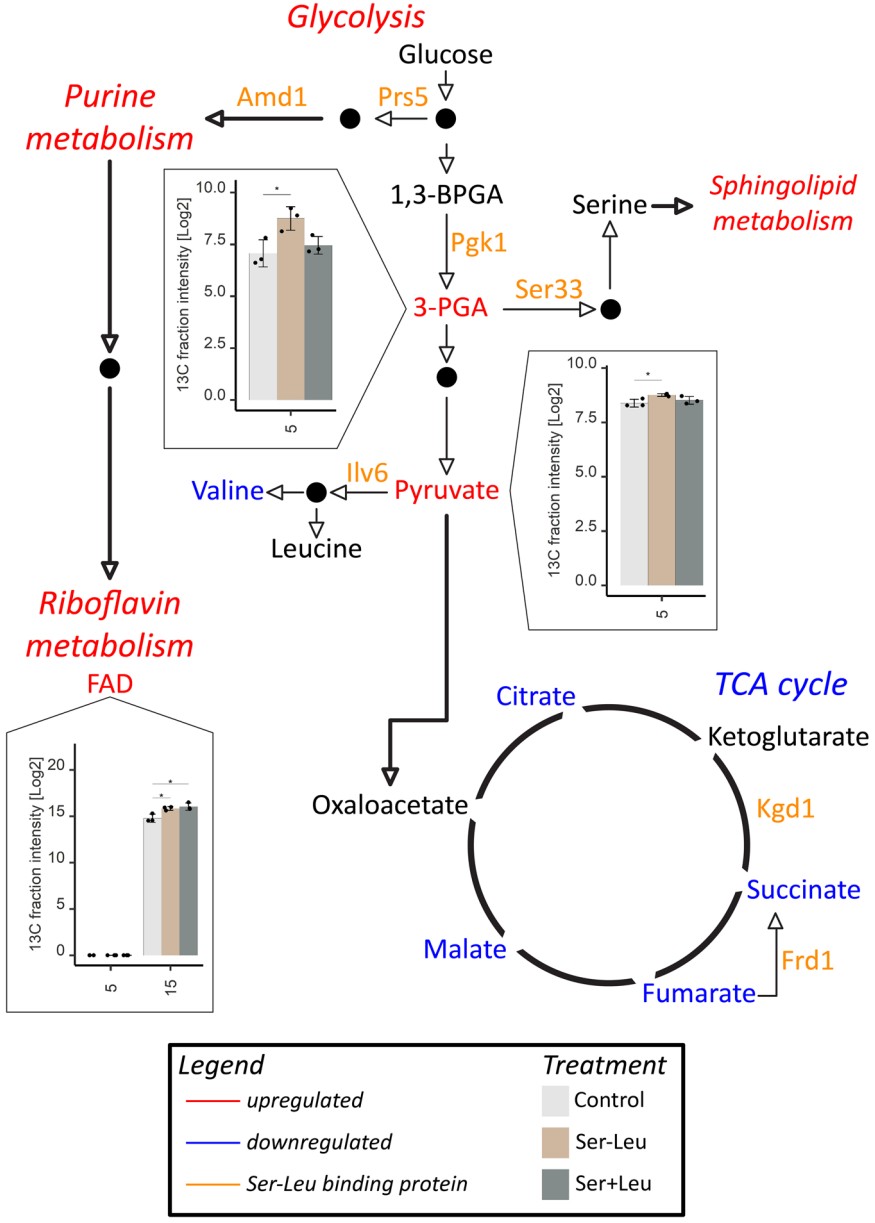

**Fig. 7 YSBN2 response to Ser-Leu supplementation.** Liquid chromatography- and gas chromatography-mass spectrometry analysis of metabolomic changes caused upon supplementation with 100 µM Ser-Leu or mixture of 100 µM serine and 100 µM leucine (Supplementary Data S32 and S33). Presented are changes in metabolite levels (here described as total intensity) and redistribution of carbon isotope (enrichment level [%] multiplied by metabolite level, here described as 13 C fraction intensity) in yeast cells. $^{13}$C enrichment in combination with metabolite levels provides information regarding the conversion rate of labelled substrate to the metabolite. X-axis represents time [min] upon treatment. Data represent the means ± SD, $n = 3$ independent samples. Asterisks denote significant difference (Tukey's test, *$P$ value < 0.05, **$P$ value < 0.01). Ser-Leu-binding proteins are marked orange. Level of metabolites marked red was significantly increased at least in one time-point comparing to other treatment. Level of metabolites marked blue was significantly decreased at least in one time point comparing to other treatment. Presented are cropped images (see Supplementary Figs. S14–S21).

The role of Pgk1 in the coordination of glycolysis and tricarboxylic acid cycle by increasing lactate production and suppressing mitochondrial pyruvate utilisation is well established in cancer cells[50–52]. Herein, and based on the similarities of yeast and cancer metabolism, we speculate that in addition to the posttranslational modifications of the mammalian Pgk1, such as phosphorylation and O-GlcNAcylation, that promote the switch from the tricarboxylic acid cycle, into lactate production, dipeptide binding may constitute an additional regulatory mechanism to promote the glycolytic capacity of cancer cells[48,53]. Notably, and in addition to being an enzyme, Pgk1 is also a protein kinase; known phosphorylation targets include pyruvate dehydrogenase kinase 1 (Pdhk1) and autophagy regulator Beclin1[52,54]. Considering that Ser-Leu increases the Pgk1 affinity towards ATP, it will be interesting to test whether Ser-Leu binding, in addition to enzymatic, affects Pgk1 kinase activity.

Although, in the present study, we focused on the Ser-Leu regulation of Pgk1; it has to be noted that Ser-Leu protein interactome comprises numerous other enzymes involved in amino acid, purine and NAD metabolism. Therefore, it is highly plausible that metabolic changes associated with Ser-Leu supplementation go beyond Pgk1 activation. For instance, Ser-Leu feeding inhibited valine production, despite the increased availability of pyruvate, which serves as a direct substrate for the

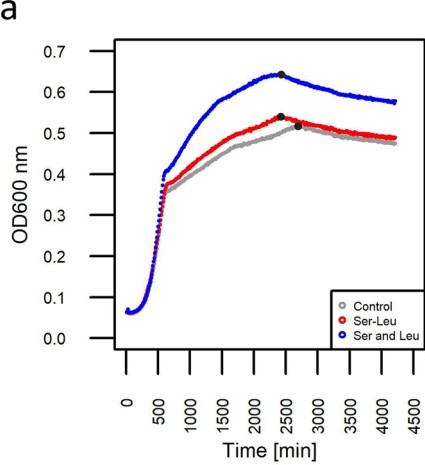

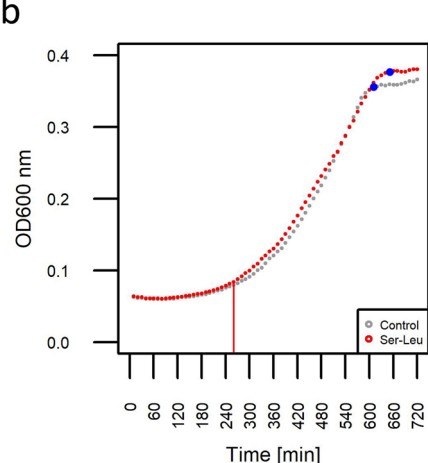

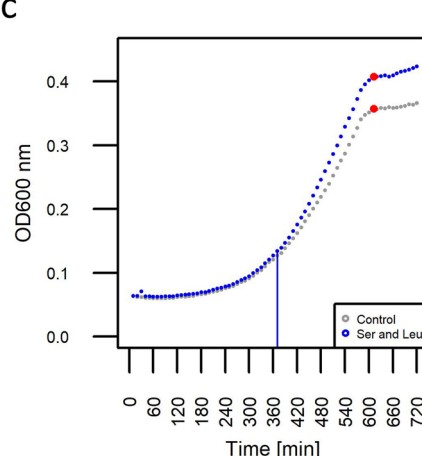

**Fig. 8 Analysis of YSBN2 growth response to Ser-Leu supplementation. a** YSBN2 strain was supplemented with 1 mM Ser-Leu or mixture of 1 mM serine and 1 mM leucine (Supplementary Data S31). Growth was monitored by measuring the optical density at 600 nm wavelength for 75 h using an automatically recording incubator. Data represent the means $n = 3$. Black dot represents end of respiration phase. **b** Cropped growth curve of YSBN2 supplemented with Ser-Leu (0–12 h). Blue dot represents diauxic shift. Red, straight line indicates beginning of Ser-Leu treatment effect on yeast growth. **c** Cropped growth curve of YSBN2 supplemented with Ser and Leu (0–12 h). Red dot represents diauxic shift. Blue, straight line indicates beginning of Ser and Leu treatment effect on yeast growth.

synthesis of branched-chain amino acids. Obtained results indicate the presence of a regulatory interaction stopping excess of pyruvate from being directed towards the synthesis of valine. We hypothesize that such regulation may be achieved by Ser-Leu inhibition of the regulatory subunit of the acetolactate synthase complex, Ilv6, which is among five high-confidence Ser-Leu protein targets. Based on the Ser-Leu co-elution with other branched-chain amino acid containing dipeptides, we also speculate that the function of Ser-Leu will be redundant with chemically similar dipeptides, and possibly even tripeptides, but as shown before for other dipeptides[55,56], different from the constituent amino acids, serine and leucine.

Finally, the regulatory role of dipeptides would become particularly important in conditions that promote protein degradation. We have recently shown that in response to abiotic stress, such as heat and dark, plants accumulate dipeptides in the autophagy-dependent manner[57]. Autophagy was also shown to account for the increase in dipeptides reported in the mammalian pro-tumorigenic cell lines[48]. Here, we could demonstrate that yeast accumulates dipeptides in response to glucose deprivation. However, it requires to be experimentally tested whether observed accumulation is autophagy dependent, glucose depletion was shown to trigger autophagy[58,59] and metabolic phenotype associated with Ser-Leu feeding such as accumulation of RNA degradation products, changes in lipid metabolism and cofactor production is reminiscent with the metabolic alterations downstream of autophagy[58,60–63].

In summary, the proteome and metabolome-wide map of the protein–protein and protein–metabolite complexes that we present here can be mined for regulatory small molecules, such as the here characterized proteogenic dipeptide Ser-Leu. Yeast growth strictly depends on the carbon availability; glucose being the primary carbon source[64]. The transition between growth on glucose to growth on ethanol is accompanied by acute metabolic rearrangement[65]. However, intensively studied, the underlying regulatory mechanisms are not entirely understood. Our work points to the involvement of proteogenic dipeptides in the control of yeast metabolism and diauxic shift, by direct regulation of enzyme activities and carbon flux. In a broader sense, presented data support proteogenic dipeptides' regulatory role at the nexus of protein degradation and central metabolism.

## Methods

**Yeast growth conditions, cell lysis and extraction of native complexes for PROMIS**. The YSBN2 strain of *S. cerevisiae* was cultivated at 28 °C with moderate shaking until it reached the logarithmic phase (OD600 = 0.3–0.5) and used for the preparation of soluble fraction containing endogenous complexes (Supplementary Methods).

**Size exclusion chromatography**. 2 mL of concentrated soluble fraction, corresponding to 40 mg of protein, was separated using a Sepax SRT SEC-300 21.2 × 300 mm column (Sepax Technologies, Inc., Delaware Technology Park, separation range 1.2 mDa to 10 kDa) connected to an AKTA explorer 10 (GE Healthcare Life Science, Little Chalfont, UK) using a 7 mL/min flow rate, 4 °C. Equilibration of the column and separation were performed using 50 mM AmBIC pH 7.5, 150 mM NaCl, 1.5 mM MgCl$_2$ and 48 1-mL fractions were collected from the 39 to 86 mL elution volume. When compared with previous studies, the separation time decreased to less than 20 min[66]. The fractions were frozen by snap freezing in liquid nitrogen and subsequently lyophilised and stored at –80 °C for metabolite and protein extractions.

The chromatogram of the absorption at 280 nm indicates reproducible fractionation ($R_{avg} = 0.98$) of the native complexes present in the input samples (Supplementary Fig. S22). To correct for unspecific metabolite binding to the column matrix, a control experiment with a protein-free sample was performed. For this purpose, proteins were precipitated from the extract of native complexes using 80% acetone. An extract of total small molecules (bound and unbound) was then solubilised in a lysis buffer before fractionation on the SEC column. The mass features present in the SEC mobile phase (blank sample) were also quantified and filtered out as potential contaminants coming from chemicals.

**Extraction of proteins and polar metabolites**. Proteins and metabolites from the lyophilised fractions were extracted using a methyl tert-butyl ether (MTBE)/methanol/water solvent system, which separates molecules into pellets (proteins), organics (lipids) and an aqueous phase (primary and secondary metabolites)[67]. Molecules were extracted from each fraction by adding 1 mL of a homogenous mixture of −20 °C methanol:MTBE:water (1:3:1), shaking for 10 min at 4 °C, incubating 10 min in an ice cooled ultrasonication bath and shaking again for 10 min at 4 °C. Next, 500 μL of UPLC grade methanol:water (1:3) was added to each fraction. The homogenates were vortexed and centrifuged for 5 min at 20,800 g, RT. Equal volumes of the polar fraction and protein pellet were dried in a centrifugal evaporator and stored at –80 °C until they were processed further. Qualitative and quantitative analysis of the fractionated proteins using the Bradford assay[68] and SDS-PAGE, respectively, showed that the majority of the proteins eluted in fractions corresponding to MW above 20 kDa (A6–C13, referred to as protein-containing fractions). Fractions C14–C15 contained low protein amounts with MWs below 20 kDa. Therefore, fractions C14 to D9 were considered to contain mostly protein fragments and metabolites that were not bound to proteins.

**LC-MS metabolomics**. After extraction, the dried aqueous phase was suspended in 100 μL of water and sonicated for 5 min using ultrasonication bath. Samples were centrifuged 10 min at 20,800 g, RT. Supernatant was transferred to UPLC glass vial. Polar metabolite extract was separated using a UPLC equipped with an HSS T3 C18 reversed-phase column and mass spectra were acquired using an Exactive mass spectrometer in positive and negative ionisation modes[67]. 3 μl of the sample was loaded onto the column for each ionisation mode. To create the required gradient for metabolite measurement, mobile phase solutions were prepared as follows: buffer A (0.1% formic acid in H$_2$O) and buffer B (0.1% formic acid in ACN). Metabolites were separated at 400 μl/min using the following gradient: 1 min 1% LC-MS mobile phase buffer B, 11 min linear gradient from 1% to 40% buffer B, 13 min linear gradient from 40% to 70% buffer B, then 15 min linear gradient from 70% to 99% buffer B, and hold a 99% buffer B concentration until 16 min. Starting from 17 min, use a linear gradient from 99% to 1% buffer B. Re-equilibrate the column for 3 min with 1% buffer B before measuring the next sample. Mass spectra were acquired using following settings: mass range from 100 to 1500 m/z, resolution set to 25,000, loading time restricted to 100 ms, AGC target set to 1e$^6$, capillary voltage to 3 kV with a sheath gas flow and auxiliary gas value of 60 and 20, respectively. The capillary temperature was set to 250 °C and skimmer voltage to 25 V.

**LC-MS/MS of proteins**. Proteins from each fraction were digested using LysC/Trypsin Mix (Promega Corp., Fitchburg, WI) according to the manufacturer's instructions. Digested proteins were desalted on self-made C18 Empore® extraction discs (3 M, Maplewood, MN) STAGE tips[69]. Dried peptides were separated using C18 reversed-phase column connected to an ACQUITY UPLC M-Class system in a 120 min gradient (Supplementary Methods).

**Data processing of LC-MS metabolite and protein data**. Data were processed using Expressionist Refiner MS 11.0 (Genedata AG, Basel, Switzerland) using settings described previously[66], with minor changes, and MaxQuant version 1.6.0.16[70] and its built-in search engine, Andromeda[71]. Detailed settings and further data processing leading to the determination of molecular complexes were described in Supplementary Methods. Additional information is also given in Supplementary Figs. S23 and S24.

**Overexpression and purification of Pnp1 and Pgk1**. Pnp1 and Pgk1 overexpressing yeast strains were purchased from Dharmacon and are part of the yeast ORF collection[44]. Yeast cultivation and procedure of protein purification were described in Supplementary Methods. Additional information is also given in Supplementary Figs. S25 and S26.

**Pnp1 enzymatic assay**. The method for Pnp1 enzymatic activity measurement was adapted from previous studies[34] (Supplementary Methods).

**Affinity purification using Ser-Leu agarose beads**. Yeast cultivation and procedure of affinity purification were described in Supplementary Methods.

**Thermal proteome profiling of the Ser-Leu-treated cell extracts**. Thermal proteome profiling of Ser-Leu-treated cell extracts was performed as described earlier[16] and analysed using a TPP package available on Bioconductor and NPARC[36] (Supplementary Methods).

**Microscale thermophoresis**. Microscale thermophoresis measurements were performed using a Monolith NT.115 instrument (Nanotemper) (Supplementary Methods).

**Pgk1 enzymatic assay**. Pgk1 activity was assayed using an optimised stopped assay, and the product was determined by an enzyme-cycling system, as described earlier, with minor modifications[45,46] (Supplementary Methods).

**Dipeptide and amino acid accumulation during growth**. Yeast was cultivated at control conditions as described earlier[72] (Supplementary Methods).

**Changes in growth and metabolism upon Ser-Leu supplementation (13C-isotope-labelling experiment)**. Chemical treatment was applied by supplementing yeast culture with either mock, 100 μM Ser-Leu or a mixture of 100 μM serine and 100 μM leucine (Supplementary Methods).

**Statistics and reproducibility**. Statistical analysis was performed using R[73]. One-way analysis of variance (ANOVA) followed by Tukey's multiple comparison test or unpaired, two-tailed Student's t test, was performed. The $P$ values < 0.05 were considered significant and are represented as *$P < 0.05$, **$P < 0.01$. For all statistical analysis data from at least three independent measurements was used. The exact number of replicates and detailed description of statistics performed are indicated in individual figure captions and methods.

## Data availability

The mass spectrometry proteomics data that support the findings of this study have been deposited in the ProteomeXchange Consortium via the PRIDE[74] partner repository with the dataset identifier PXD021530. Source data underlying figures are available in Supplementary Data S1–S33. All other data are available from the corresponding author on reasonable request.

## Code availability

R code used for data processing and analysis was submitted to GitHub repository and deposited to Zenodo[75]. Code can be accessed at https://github.com/Marcin-Luzarowski/PROMIS.git or https://doi.org/10.5281/zenodo.4146637.

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

## Acknowledgements

We thank Änne Michaelis and Fatıma Şen for excellent technical assistance; Krzysztof Bajdzienko, Dariusz Bienkowski and Andreas Donath for help with establishing a user-friendly interface; and Prof. Dr. Zoran Nikoloski for a careful read of the manuscript and valuable comments.

## Author contributions

M.L. and A.S. devised the experimental strategy and wrote the manuscript; M.L. supervised the work of undergraduate students; J.E. and A.R.F helped with writing the manuscript and reviewed the data; M.L., A.K., R.V., I.W. and M.W. executed the experiments; M.L., R.V., U.L., A.E. and D.C. analysed the data; D.S. created user interface; M.G. created the deconvolution script; M.G. and E.S. assisted with proteomics measurements; M.K assisted with the TPP experiment; J.M., D.S., L.P.S. and J.K. provided valuable discussion and assisted in data analysis; A.B. and B.V. assisted with protein purification; A.S. and L.W. coordinated the project.

## Funding

## Competing interests

The authors declare no competing interests.

## Additional information

[1]Department of Molecular Physiology, Max Planck Institute of Molecular Plant Physiology, Potsdam, Germany. [2]Department of Metabolic Networks, Max Planck Institute of Molecular Plant Physiology, Potsdam, Germany. [3]Laboratoire de Recherche en Sciences Végétales (LRSV), UPS/CNRS, UMR, Castanet Tolosan, France. [4]University of Wrocław, Faculty of Biotechnology, Laboratory of Medical Biology, Wrocław, Poland. [5]Department of Genome Biology, European Molecular Biology Laboratory, Heidelberg, Germany. [6]Department of Life Sciences, Ben-Gurion University of the Negev, Beer-Sheva, Israel. [7]Center for Desert Agriculture, Biological and Environmental Science and Engineering Division (BESE), King Abdullah University of Science and Technology (KAUST), Thuwal, Saudi Arabia. [8]Interfaculty Institute of Cell Biology, Eberhard Karls University of Tuebingen, Tuebingen, Germany. [9]Boyce Thompson Institute, Ithaca, NY, USA. ✉email: luzarowski@mpimp-golm.mpg.de; skirycz@mpimp-golm.mpg.de

