## [Peer Review File · Communications Biology]

Reviewers' comments:

Reviewer #1 (Remarks to the Author):

General description

The authors have developed a technique named PROMIS to globally explore protein-metabolite interactions, something that other techniques cannot do without baiting specific interactions. They confirm masses of coelution peaks with public databases and already known complexes such as the proteasome and they capture previously predicted interactions.

The authors point to an interesting interaction they detected between Pnp1 and xanthine which they validated with enzymatic assays testing the activity of the enzyme after the addition of xanthine, unfortunately in only one concentration (100 μ M). They have the hypothesis that GUD1 that degrades guanine leads subsequently to the production of xanthine that leads to the proliferation to quiescence. No further experiments were done or the expression of GUD1 was checked in specific time-points. Then they focused on dipeptides and Ser-Leu had 94 potential targets with an interesting enrichment in metabolism. Also, they identified a coelution of Ser-Leu with Pgc1 which was experimentally validated. The dipeptide increases the activity of Pgc1 but ATP presence reduces this effect. With growth experiments, they showed that the dipeptide accumulates when glucose is depleted by the medium. Ser-Leu treatment directs metabolism away from the TCA cycle with the intermediates being decreased, as shown by ¹³C metabolomics experiments. It also affected growth when supplemented, with delaying the diauxic shift for 30 minutes with the fermentation phase getting extended.

Comments

It is a work that was constructed on the abilities of the PROMIS method and revealed the importance of global screening. Xanthine - Pnp1 interaction was discovered and the role of dipeptides, particularly Ser-Leu in metabolism, was revealed. In most cases, the authors validate initial findings with other experiments and come with new questions that emerge from previous findings. The role of Ser-Leu accumulation on the metabolic shift from fermentation to quiescence is important. The transition from PROMIS to xanthine - Pnp1 and then to Ser-Leu is not so smooth and it would be better to focus more in the Ser - Leu, as the title of the manuscript indicates. The hypothesis they make on the xanthine role is not experimentally tested. In the discussion part, the authors claim that they have mechanistically understood the dipeptide regulation, but this is not the case as more experiments would be needed (i.e. inhibition or knock out of particular genes of interest and rescue experiments). So the experiments seem to have been executed in high level, the results are important considering the value of the screening strategy they have developed and the revealed important role of dipeptides (Ser-Leu) but more should be done in the biological interpretation, although there is a very good start with the tracing metabolomics profiling.

Specific comments

- The screening technology is exciting and should be published
- Growth experiments and mechanistic examination could be performed before the hypothesis for GUD1 is considered as a mature one
- Figure 4C is a result but does not match with the story as no further attention is paid in the manuscript apart from a small paragraph
- In Figure 6C the leucine growth profile seems similar to Ser-Leu for the first 60 minutes. Any ideas why?
- Reference 5 in the methods section needs correction. It is the R software reference
- It would be nice for the authors to provide data on how they ended up with 6 μ M Ser-Leu spiking
- Further discussion about the metabolomics results would be a good idea
- Authors discuss potential autophagy mechanism but they do not provide experimental data

Reviewer #2 (Remarks to the Author):

In "Dipeptide Ser-Leu acts as metabolic switch at the interface of protein and central metabolism.", Luzarowski et al present a new approach for PMI detection, PROMIS, and observe experimentally several predicted PMIs. The authors then validate interesting PMIs using many different approaches. Overall, I found the manuscript to be well written and interesting. The method offers the opportunities for new biological insights and should be of interest for researchers both in the proteomics and metabolomics field. I was impressed by the amount of work that was put in the validation studies. I think the experimental approach is sound, but I have some concerns regarding the statistical analysis of data in the PROMIS pipeline. Some important information and metrics are missing from the results and the methods.

1. First, I am unsure that the Pearson Correlation Coefficient is the best approach to measure the correlation between elution profiles after deconvolution. From what I understood in the methods, most of the values in the elution profile will be close to 0 or input as missing values. I think a the kendall-tau correlation, which is non-parametric and is better equipped to handle ties, would be more appropriate for this type of analysis.
2. There should also be a figure showing the overall distribution of correlation coefficients for all peak pairs which shows where the cut-off selected by the authors fall. The author should also provide the False Discovery Rate for the different correlation coefficient cut-off.
3. Figure S1 compares the ranked mean correlation of 100 permuted datasets to the set of "True" positives extracted from the STITCH database. Instead of representing only the mean, the authors should also show the standard deviation around the mean.
4. Figure 2 bars should be split in three categories: previously reported, predicted and this experiment only.
5. I found concerning the fact that the positive rate was higher (225/1122, ~20%) for the predicted PMIs set compared to the true positive set (14/87, ~16%). This suggests either that the true positive set is biased for PMIs that are not well detected by PROMIS, or that the correlation coefficient cut-off is too permissive and that some of these hits are false positives. Could the authors show the number of predicted PMIs found as a function of the stringency of the STRING interaction score? Are there more observed predicted PMIs with high STRING scores?
6. It was also unclear to me how PMIs with multiple peaks above the correlation coefficient threshold (such as PNP1-inosine in figure 3) are handled. Intuitively, multiple correlated peaks should indicate a PMI with higher confidence: what would be the chance of such a thing happening in the permuted dataset?
7. The methods lack details on how data was analyzed and how figures were generated: scripts should be provided either on github or as supplementary files. As a large chunk of the paper is the PROMIS method, and as such the scripts and tools used for data analysis should be easily accessible.

Some minor comments

8. On page 6:

Twenty-seven percent of the identified proteins were annotated as subunits of protein complexes, 7.5% were involved in molecular transport, 5% were kinases, and 8% had putative or unknown functions. Proteins integral to membranes or associated with the plasma membrane were significantly underrepresented (0.71-fold enrichment), whereas cytoplasmic and nucleolar proteins were overrepresented (1.28 and 1.48-fold enrichment, respectively) (PANTHER database, Table S4). Please provide values for each category across the proteome to make clear whether or not this is expected.

9. On page 9:

To validate the predicted partners, we performed affinity purification (AP) experiments starting with Ser-Leu as the bait (c.f. Materials and Methods). We used agarose beads coupled to Ser-Leu by the

NH₂ group of serine (Ser-Leu-N) or the COOH group of leucine (Ser-Leu-C).
I think renaming Ser-Leu-N to N-Ser-Leu would make the text clearer to the reader.

10. Many panels of figure S6 display datapoints outside the plot area.

Reviewer #1

We are grateful for the reviewer's comments/suggestions. We addressed them as follows:

General Comments

It is a work that was constructed on the abilities of the PROMIS method and revealed the importance of global screening. Xanthine - Pnp1 interaction was discovered and the role of dipeptides, particularly Ser-Leu in metabolism, was revealed. In most cases, the authors validate initial findings with other experiments and come with new questions that emerge from previous findings. The role of Ser-Leu accumulation on the metabolic shift from fermentation to quiescence is important.

The transition from PROMIS to xanthine - Pnp1 and then to Ser-Leu is not so smooth and it would be better to focus more in the Ser - Leu, as the title of the manuscript indicates. The hypothesis they make on the xanthine role is not experimentally tested.

We agree with the reviewer that the hypothesis we put forward regarding xanthine's role in the regulation of nucleotide metabolism needs to be experimentally tested. However, considering the 1) focus of the current manuscript, 2) the amount of experimental work it would require 3), and the amount of validation work that is already present, we feel it is more suitable for a future manuscript.

We chose the interaction between Pnp1 and xanthine as an example where mining PROMIS dataset alone is sufficient to identify meaningful interactions, and we proved that by testing Pnp1 enzymatic activity in the presence of xanthine. To address the reviewer's criticism, we revised the results and discussion, clearly stating that the role of xanthine in the regulation of nucleotide metabolism remains a hypothesis.

In the discussion part, the authors claim that they have mechanistically understood the dipeptide regulation, but this is not the case as more experiments would be needed (i.e. inhibition or knock out of particular genes of interest and rescue experiments). So the experiments seem to have been executed in high level, the results are important considering the value of the screening strategy they have developed and the revealed important role of dipeptides (Ser-Leu) but more should be done in the biological interpretation, although there is a very good start with the tracing metabolomics profiling.

We agree with the reviewer about the value of the additional genetic evidence for a detailed, mechanistic understanding of the biological role of Ser-Leu. Ideally, phenotypic and metabolic characterization of the *pgk1* mutant strain complemented with the Pgp1 defective in dipeptide binding but unaffected in its enzymatic activity. However, considering the amount of experimental work it would require (structural analysis to uncover Ser-Leu regulatory site, followed by biochemical characterization of wild-type and mutant proteins, strain complementation and metabolic characterization), we feel it is more suitable for a future manuscript.

To address the reviewer's criticism, we moderated our statement about the mechanistic understanding of Ser-Leu regulation (see Discussion).

Specific Comments:

Comment 1: The screening technology is exciting and should be published.

We thank you, Reviewer, for the supportive comment.

Comment 2: Growth experiments and mechanistic examination could be performed before the hypothesis for GUD1 is considered as a mature one

Please see our response above.

Comment 3: Figure 4C is a result but does not match with the story as no further attention is paid in the manuscript apart from a small paragraph.

Thank you for pointing this out. To address the reviewer's criticism, we now address a putative role of Ser-Leu (and other branched-chain amino-acids) in valine biosynthesis in the Discussion.

Comment 4: In Figure 6C the leucine growth profile seems similar to Ser-Leu for the first 60 minutes. Any ideas why?

Recovery from the stationary phase is associated with extensive metabolic rewiring. We don't have an explanation of why both Ser-Leu and leucine decrease over the first 60 minutes. The aim of this figure was to show that Ser-Leu accumulates when glucose is consumed from the medium, and that accumulation of dipeptides is independent of changes in the amino acid levels. We replotted Figure 4C to highlight that Ser-Leu accumulates several hours before leucine.

Comment 5: Reference 5 in the methods section needs correction. It is the R software reference.

We updated the reference list by updating the R software reference and adding R language reference.

Comment 6: It would be nice for the authors to provide data on how they ended up with 6 μ M Ser-Leu spiking.

Requested data can now be found in the new Table S26.

Comment 7: Further discussion about the metabolomics results would be a good idea

As requested, we introduced additional discussion regarding metabolomics results in both the results and discussion sections.

Comment 8: Authors discuss potential autophagy mechanism but they do not provide experimental data

To address the reviewer's comment, we toned down our claims regarding autophagy. We refer to the findings from the plant and mammalian field, but only as means of discussion.

Reviewer #2

We are grateful for the reviewer's comments/suggestions. We addressed them as follows:

Comment 1: First, I am unsure that the Pearson Correlation Coefficient is the best approach to measure the correlation between elution profiles after deconvolution. From what I understood in the methods, most of the values in the elution profile will be close to 0 or input as missing values. I think a the kendall-tau correlation, which is non-parametric and is better equipped to handle ties, would be more appropriate for this type of analysis.

We thank you, reviewer, for his/her suggestion. We decided on the Pearson correlation, as it succeeded in enriching for known and identifying the new interactions before (Veyel et al., 2018). It also worked well for the current dataset. Re-analysing the dataset would require that we make significant changes to the manuscript, such as replotting almost all the figures.

However, we are working on a user-friendly, web-based application, which will allow users to (re)-analyse PROMIS data. Based on the Reviewer's comment, we included the possibility to use Kendall-tau correlation method to predict small molecule – protein complexes. The tool for data analysis will be released as a separate method manuscript. A preview version can be found under the following link:

<https://promis.mpimp-golm.mpg.de/>

username: webreview

password: way2Judge2020!

Comment 2: There should also be a figure showing the overall distribution of correlation coefficients for all peak pairs which shows where the cut-off selected by the authors fall. The author should also provide the False Discovery Rate for the different correlation coefficient cut-off.

We provided an additional figure showing the overall distribution of correlation coefficients (Figure S27). Additionally, we provided a supplementary table showing False Discovery Rate for the different correlation coefficient cut-off (Table S10).

Comment 3. Figure S1 compares the ranked mean correlation of 100 permuted datasets to the set of "True" positives extracted from the STITCH database. Instead of representing only the mean, the authors should also show the standard deviation around the mean.

We provided new Figure S1, in which we included the SD around the mean.

Comment 4. Figure 2 bars should be split in three categories: previously reported, predicted and this experiment only.

We provided new Figure 2A, which is now split in three categories.

Comment 5: I found concerning the fact that the positive rate was higher (225/1122, ~20%) for the predicted PMIs set compared to the true positive set (14/87, ~16%). This suggests either that the true positive set is biased for PMIs that are not well detected by PROMIS, or that the correlation coefficient cut-off is too permissive and that some of these hits are false positives. Could the authors show the number of predicted PMIs found as a function of the stringency of the STRING interaction score? Are there more observed predicted PMIs with high STRING scores?

Please, find below the figure showing the Pearson correlation coefficient calculated for found peaks as a function of the stringency of the STITCH interaction score (either combined or experimental score). There is no relation between the stringency of the score and Pearson correlation and the number of captured interactions. A higher positive rate for the predicted PMIs set might be due to the presence of dozens of predicted interactions between nucleoside monophosphates and RNA binding proteins, which were captured in our experiment (please, see Figure 2A and Table S10).

Figure Legend. Pearson correlation coefficient calculated for found peaks as a function of the stringency of the STITCH interaction score (either combined or experimental score). Red color was use to mark interaction with correlation above used threshold (0.7 PCC).

Comment 6: It was also unclear to me how PMIs with multiple peaks above the correlation coefficient threshold (such as PNPI-inosine in figure 3) are handled. Intuitively, multiple correlated peaks should indicate a PMI with higher confidence: what would be the chance of such a thing happening in the permuted dataset?

We agree with the Reviewer that intuitively, multiple correlated peaks should indicate a PMI with higher confidence. However, this assumption holds if small-molecule interact with protein regardless of protein oligomeric state, which is often not the case. The results of our analysis have to be considered as

rather “qualitative” than “quantitative”. We do not claim that a higher Pearson correlation indicates the occurrence of the interaction with higher probability. We did not assess this problem in our manuscript, and we do not provide a scoring system.

Comment 7: The methods lack details on how data was analyzed and how figures were generated: scripts should be provided either on github or as supplementary files. As a large chunk of the paper is the PROMIS method, and as such the scripts and tools used for data analysis should be easily accessible.

We are working on user-friendly, web-based application, which will allow fast analysis of PROMIS datasets. Scripts and tools will be therefore published as a method paper. To attest it, a preview version can be found under the following link:

<https://promis.mpimp-golm.mpg.de/>
username: webreview

password: way2Judge2020!

Comment 8: Twenty-seven percent of the identified proteins were annotated as subunits of protein complexes, 7.5% were involved in molecular transport, 5% were kinases, and 8% had putative or unknown functions. Proteins integral to membranes or associated with the plasma membrane were significantly underrepresented (0.71-fold enrichment), whereas cytoplasmic and nucleolar proteins were overrepresented (1.28 and 1.48-fold enrichment, respectively) (PANTHER database, Table S4). Please provide values for each category across the proteome to make clear whether or not this is expected.

Data are available in Supplementary Table S4 (Column B, Row 9). Moreover, we added values for each category in the main text.

Comment 9: To validate the predicted partners, we performed affinity purification (AP) experiments starting with Ser-Leu as the bait (c.f. Materials and Methods). We used agarose beads coupled to Ser-Leu by the NH₂ group of serine (Ser-Leu-N) or the COOH group of leucine (Ser-Leu-C). I think renaming Ser-Leu-N to N-Ser-Leu would make the text clearer to the reader.

We applied suggested changes to the manuscript.

Comment 10: Many panels of figure S6 display data-points outside the plot area.

We believe Reviewer 2 meant Figure S8 showing melting profiles of 94 putative targets of Ser-Leu. Is this correct? We would like to kindly notice that some points overlap with the legend but are not outside the plot area. We hope the figure is acceptable in the present form.

Reviewers' comments:

Reviewer #1 (Remarks to the Author):

The authors have addressed most comments of Reviewer #1, and some of Reviewer #1. But in turn, Reviewe #1 does not asked any thing very concrete, other than hoping for the authors could provide a better mechanistic understanding, which I understand, is not that simple to provide. From my end, I think this paper is now a candidate for being published.

Reviewer #2 (Remarks to the Author):

I have read the revised version of the article and the answer of the authors to comments. I thank the authors for taking the time to make most of the requested changes and perform some additional analysis steps as well. However, I found that some of my concerns were not fully addressed.

1. First, the authors did not perform any additional analysis to address my concerns on statistical testing using Pearson correlation instead of Kendall-Tau, citing two reasons. First, they point to the original PROMIS article (Veyel et al., 2018), stating: "We decided on the Pearson correlation, as it succeeded in enriching for known and identifying the new interactions before (Veyel et al., 2018). It also worked well for the current dataset". Having also read Veyel et al 2018, I found no data demonstrating that Pearson correlation was benchmarked against other statistical tests or data analysis approaches. Thus, it cannot serve as justification for not performing the additional analysis work.

Second, the authors state "Re-analysing the dataset would require that we make significant changes to the manuscript, such as replotting almost all the figures" in their rebuttal letter. If the authors had performed additional analysis work showing that their approach performed well with both tests, I would agree that there is probably no need to change all the figures. I understand this could entail a significant amount of work, but I if it ends up making the method more robust it will have been worth it.

2. Table S10 provides the true positive (TPR) and false positive rate (FPR), but not the false discovery rate (FDR). The false discovery rate is the fraction of positives expected to be false hits ($FDR = FPR / (TPR + FPR)$). At the current PCC threshold chosen by the authors, the $FDR = 17.6\%$ ($3.45 / (16.092 + 3.45)$, values from table S10). This is a bit high, and should definitely be mentioned in the main text as it suggests that almost a fifth of PMIs detected are not true hits. The way the false positive rate is measured should also be clarified in the main text and the methods, as is remains unclear in the revised manuscript.

6. I thank the authors for their explanation. However, I think that if the approach provides a qualitative result, statements such as "We provide a unique data set of 225 high-confidence interactions" should be moderated to reflect that it is in fact not feasible to say which interactions have a higher confidence than others in the set. Furthermore, this should change the interpretation of the results from shown in figure S1, where instead of looking at the PCC value distribution what should be measured is the number of random pairs with a $PCC > 0.7$ per permutation over a large number of iterations. This would allow for a better estimation of the false positive rate, which could then be used to evaluate the false discovery rate for the threshold PCC.

7. While I appreciate that a web app for PROMIS data analysis is in development, this does not justify not including the code used for data analysis in this manuscript in my opinion. Depending on the time required for publication of the bioinformatics method paper, there might be a window of months to over a year where the software used to analyze the data is not available to researchers that would want to reproduce the results of this study. I also fail to see how providing this information here would detract from the methods paper and the dedicated web server that will be made available to the community.

Reviewer #1

We would like to thank reviewer for accepting our work for publication.

Reviewer #2

We are grateful for the reviewer's comments/suggestions. We addressed them as follows:

First, the authors did not perform any additional analysis to address my concerns on statistical testing using Pearson correlation instead of Kendall-Tau, citing two reasons. First, they point to the original PROMIS article (Veyel et al., 2018), stating: "We decided on the Pearson correlation, as it succeeded in enriching for known and identifying the new interactions before (Veyel et al., 2018). It also worked well for the current dataset". Having also read Veyel et al 2018, I found no data demonstrating that Pearson correlation was benchmarked against other statistical tests or data analysis approaches. Thus, it cannot serve as justification for not performing the additional analysis work. Second, the authors state "Re-analysing the dataset would require that we make significant changes to the manuscript, such as replotting almost all the figures" in their rebuttal letter. If the authors had performed additional analysis work showing that their approach performed well with both tests, I would agree that there is probably no need to change all the figures. I understand this could entail a significant amount of work, but I if it ends up making the method more robust it will have been worth it.

We thank you, reviewer, for his/her suggestion. As requested, we now compared performance of Pearson correlation and Kendall-Tau. The resulting receiver operating characteristic curve is comparable for both approaches; therefore we decided to stay with the Pearson correlation. However, and as stated before, we find the suggestion very valuable and we included Kendall-Tau Correlation in the PROMIS App, we are planning to release next year.

Figure 1: ROC curve prepared using known PM complexes retrieved from Stitch: Blue – Pearson Correlation, Red – Kendall-Tau Correlation

Table S10 provides the true positive (TPR) and false positive rate (FPR), but not the false discovery rate (FDR). The false discovery rate is the fraction of positives expected to be false hits ($FDR = FPR / (TPR + FPR)$). At the current PCC threshold chosen by the authors, the $FDR = 17.6\%$ ($3.45 / (16.092 + 3.45)$, values from table S10). This is a bit high, and should definitely be mentioned in the main text as it suggests that almost a fifth of PMIs detected are not true hits. The way the false positive rate is measured should also be clarified in the main text and the methods, as it remains unclear in the revised manuscript.

False discovery rate is now provided in Table S10 and its value is reported in the main text (see Results). In the supplementary materials and methods, and results we clarified that false positive rate was determined by randomly picking PCC for any given protein-metabolite pair.

I thank the authors for their explanation. However, I think that if the approach provides a qualitative result, statements such as “We provide a unique data set of 225 high-confidence interactions” should be moderated to reflect that it is in fact not feasible to say which interactions have a higher confidence than others in the set.

As suggested the statement was moderated. We also explain qualitative nature of the method in the results section.

Furthermore, this should change the interpretation of the results from shown in figure S1, where instead of looking at the PCC value distribution what should be measured is the number of random pairs with a $PCC > 0.7$ per permutation over a large number of iterations. This would allow for a better estimation of the false positive rate, which could then be used to evaluate the false discovery rate for the threshold PCC.

Following reviewer’s suggestion we provide additional information in Table S10 regarding the number of random pairs with a PCC above given threshold per permutation over 100 iterations. Based on this number, we calculated alternative False Positive Rate and False Discovery Rate for any given PCC threshold in a range from 0 to 1. 0.7 PCC threshold is in the top ten PCC thresholds granting the lowest FDR. Below, please find a figure showing distribution of randomly picked PCC over 100 iterations. Red, horizontal line represents 0.7 PCC threshold used in this study. Both approaches used to determine false discovery rate point to 0.7 PCC being an optimal threshold for selecting interactors.

While I appreciate that a web app for PROMIS data analysis is in development, this does not justify not including the code used for data analysis in this manuscript in my opinion. Depending on the time required for publication of the bioinformatics method paper, there might be a window of months to over a year where the software used to analyze the data is not available to researchers that would want to reproduce the results of this study. I also fail to see how providing this information here would detract from the methods paper and the dedicated web server that will be made available to the community.

The code used for data analysis is now publicly available (please see, <https://github.com/Marcin-Luzarowski/PROMIS.git>)

Reviewer #2 (Remarks to the Author):

I thank the authors for taking the time to answer my comments and make the requested changes to the manuscript. As all the points I raised have been thoroughly addressed.